# A comprehensive benchmarking with practical guidelines for cellular deconvolution of spatial transcriptomics

Haoyang Li [1,2,6], Juexiao Zhou[1,2,6], Zhongxiao Li [1,2], Siyuan Chen [1,2], Xingyu Liao [1,2], Bin Zhang[1,2], Ruochi Zhang[3], Yu Wang[3], Shiwei Sun[4,5] & Xin Gao [1,2] ✉

Spatial transcriptomics technologies are used to profile transcriptomes while preserving spatial information, which enables high-resolution characterization of transcriptional patterns and reconstruction of tissue architecture. Due to the existence of low-resolution spots in recent spatial transcriptomics technologies, uncovering cellular heterogeneity is crucial for disentangling the spatial patterns of cell types, and many related methods have been proposed. Here, we benchmark 18 existing methods resolving a cellular deconvolution task with 50 real-world and simulated datasets by evaluating the accuracy, robustness, and usability of the methods. We compare these methods comprehensively using different metrics, resolutions, spatial transcriptomics technologies, spot numbers, and gene numbers. In terms of performance, CARD, Cell2location, and Tangram are the best methods for conducting the cellular deconvolution task. To refine our comparative results, we provide decision-tree-style guidelines and recommendations for method selection and their additional features, which will help users easily choose the best method for fulfilling their concerns.

Spatial transcriptomics technologies, named "Method of the Year 2020"[1], have undergone rapid development in recent years. They are used to profile spatial locations of all detected mRNAs, providing a new perspective for biologists seeking to understand cells per se as well as their microenvironments. Broadly, spatial transcriptomics technologies can identify undiscovered transcriptional patterns and reconstruct transcriptional panoramas of whole tissues. On a fine-grained level, these technologies can be used to explore the interactions among neighboring cells and intracellular and extracellular states, which helps redefine the function of cells and improves our knowledge of diseases[2]. The current spatial transcriptomics technologies can be mainly classified into two categories. The first

category is image-based technologies, including in situ sequencing- and in situ hybridization-based methods[3], which can profile mRNA with high spatial resolution, especially at the subcellular level. However, limitations such as the low number of profiled genes, low sensitivity of mRNA detection, and time-consuming processes impede the broad application of image-based technologies. The second category is sequencing-based spatial transcriptomics technologies, which capture position-barcoded mRNA with nongene-specific probes. These technologies can profile the whole transcriptome of tissue sections of any size, and are more user-friendly and less time-consuming than image-based technologies[4]. Moreover, spatial transcriptomics technologies are highly applicable and have been used to

[1]Computational Bioscience Research Center, King Abdullah University of Science and Technology (KAUST), Thuwal, Saudi Arabia. [2]Computer, Electrical and Mathematical Sciences and Engineering Division, King Abdullah University of Science and Technology (KAUST), Thuwal, Saudi Arabia. [3]Syneron Technology, Guangzhou 510000, China. [4]Key Lab of Intelligent Information Processing, Institute of Computing Technology, Chinese Academy of Sciences, 100190 Beijing, China. [5]University of Chinese Academy of Sciences, 100049 Beijing, China. [6]These authors contributed equally: Haoyang Li, Juexiao Zhou. ✉e-mail: xin.gao@kaust.edu.sa

improve our understanding of various species, organs, and tissues, including the brain[5], liver[6], and tumors[7].

One critical issue related to sequencing-based spatial transcriptomics technologies is low-resolution spots containing multiple cells with several blended cell types, which can conceal the genuine transcriptional pattern and lead to biological misunderstanding of the tissue resulting in the distorted cellular-level reconstruction of the tissue. An important task, therefore, is to quantify the proportion of all cell types among captured spots, so-called cellular deconvolution. Following deconvolution, all captured spots can be used to better understand intercellular functions and recover the fine-grained panorama of a heterogeneous tissue.

A recent benchmarking study[8] was focused on single-cell RNA sequencing (scRNA-seq) and spatial transcriptomics integration methods. There is another recent benchmarking study[9], in which the number of related methods is limited and scRNA-seq reference-free methods are not considered. Despite these efforts, clear guidelines and solid recommendations for the users are still lacking for the comprehensive coverage of available methods.

In the present study, we conducted a comprehensive benchmarking and provided guidelines for the cellular deconvolution of spatial transcriptomics data. Specifically, we evaluated 18 existing computational methods with 50 simulated and real-world datasets by comprehensively testing the accuracy, robustness, and usability of the methods. These methods could be broadly classified as those with and without scRNA-seq references. Based on their computational techniques, we grouped the methods as follows: probabilistic-based, non-negative matrix factorization-based (NMF-based), graph-based, optimal-transport (OT)-based and deep learning-based methods. During benchmarking, we used multiple metrics and various data resources with different spatial transcriptomics techniques, spot resolutions, gene numbers, spot numbers, and cell types to ensure our assessment was comprehensive and to deepen our understanding of the cellular deconvolution methods. In addition to the quantification and visualization processes, decision-tree-style guidelines were produced, which included the refinement of the benchmarking results and the collection of respective additional features of the methods detailed in related publications. These guidelines recommend scenario-specific methods for users considering computational efficiency and the characteristics of data resources. The general limitations and future perspectives associated with cellular deconvolution are also discussed to give users a clear picture of the cellular deconvolution field and thus facilitate the improvement of tools for the community.

## Results

### Benchmarking pipeline

To evaluate cellular deconvolution methods comprehensively, we identified 18 existing methods from published and preprint papers as follows: CARD[10], Cell2location[11], RCTD[12], DestVI[13], stereoscope[14], SpatialDecon[15], STRIDE[16], NMFreg[17], SpatialDWLS[18], SPOTlight[19], DSTG[20], SD2[21], Tangram[22], Berglund[23], SpiceMix[24], STdeconvolve[25], SpaOTsc[26] and novoSpaRc[27]. According to the data resources used, Berglund, SpiceMix, and STdeconvolve were scRNA-seq reference-free methods that identified cell-type-specific spatial patterns using only the information from the spatial locations of spots and their gene expression profiles without any reliance on external scRNA-seq data. The remaining 15 methods required scRNA-seq data from the same tissue as the spatial transcriptomics data. Cell-type annotations and cell-type-specific gene expression profiles from scRNA-seq data can help optimize the proportion of all cell types in the spatial transcriptomics data. The 18 methods were classified based their computational techniques as follows. Probabilistic-based methods: Berglund, Cell2location, DestVI, RCTD, SpatialDecon, stereoscope, STRIDE, and STdeconvolve; NMF-based methods: CARD, NMFreg, SpatialDWLS, SPOTlight, and SpiceMix; graph-based methods: DSTG and SD2; deep

learning-based method: Tangram; and OT-based: SpaOTsc and novoSpaRc. These five computational techniques introduced different methods to formulate the spatial transcriptomics data (sometimes with the scRNA-seq data) and solve the cellular deconvolution problem (Fig. 1A).

We also collected seven image-based and sequencing-based spatial transcriptomics datasets: seqFISH+[28], MERFISH[29,30], Spatial Transcriptomics (ST)[31], 10X Visium (Visium)[32], Slide-seqV2[33], and stereo-seq[34,35]. Their corresponding scRNA-seq datasets were collected as complementary resources (Supplementary Table 1). Among these data resources, the image-based spatial transcriptomics data (seqFISH+ and MERFISH) contained gene expression profiles, spatial locations, and cell-type annotations of individual cells, which could be used to simulate low-resolution spots by binning the cells with a unified square size, and the ground truth could be calculated according to the number of cells with different cell types in each spot. Simulated data could be used to generate different resolutions of spots by defining the different sizes of the binning squares. For the sequencing-based spatial transcriptomics data (ST, Visium, Slide-seqV2, and stereo-seq), real-world scenarios were included that could be solved by the cellular deconvolution methods. The resolution statistics and the number of genes and spots from the abovementioned data resources are plotted in Fig. 1B. Notably, the spots from stereo-seq were of a subcellular resolution (500 nm), and we binned the stereo-seq spots to reduce the resolution for cellular deconvolution (see "Methods").

After obtaining deconvolution results from all 18 methods on all spatial transcriptomics datasets, we assessed the performances of the methods comprehensively according to the following quantities (Fig. 1C): (1) the accuracy of the deconvolution results, which was evaluated using multiple metrics on all methods and datasets; (2) the robustness of all methods tested on different conditions (spatial transcriptomics techniques, number of genes, number of spots, and number of cell types); (3) the usability of all tools, including computational efficiency, quality of documents, publications, and code. To display the benchmarking results of all methods intuitively, a table containing an evaluation of all metrics according to the three listed qualities is provided (Fig. 2), in which darker dots represent better performance (see "Methods"). Moreover, detailed decision-tree-style guidelines are provided, which include scenario-specific recommendations of the methods and a summary of their respective additional features detailed in their related publications.

### Accuracy

To assess accuracy of methods, we used multiple metrics to quantify the performance of deconvolution, including the Jensen–Shannon divergence (JSD) score, root-mean-square error (RMSE), and Pearson correlation coefficient (PCC). For simulated data (MERFISH and seqFISH+), we used the JSD score and RMSE to measure the distance between the predicted cell-type proportion and ground truth (see "Methods"). Because seqFISH+ and MERFISH have single-cell resolutions, we binned them by size (51.5 μm and 100 μm, respectively) to simulate low-resolution spots. According to the number of simulated spots and genes detected for seqFISH+ and MERFISH, seqFISH+ (71 spots and 10,000 genes) had a low number of spots but a high number of genes, whereas MERFISH (3067 spots and 135 genes) exhibited the opposite trend. These two data resources were complementary in terms of the number of spots and genes, and hence provided helpful results to observe the performance of all methods at these two extremes. For each sequencing-based dataset (ST, Visium, Slide-seqV2, and stereo-seq), we chose several cell types and their known marker genes to compare the PCC between the spatial distribution of cell-type proportion and marker-gene expression, which was used to quantify the performance of deconvolution without any ground truth (Supplementary Table 2; see the "Methods").

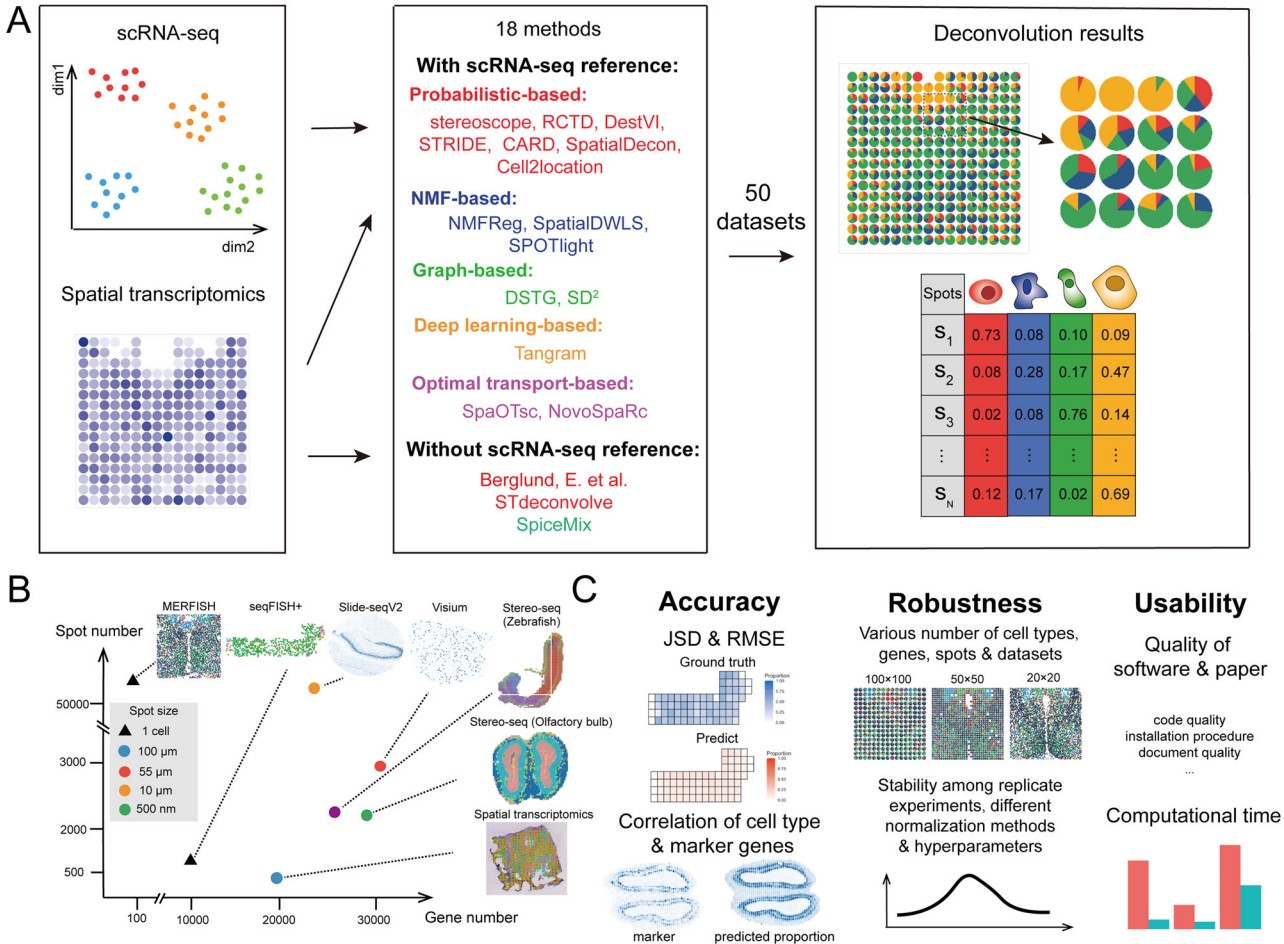

**Fig. 1 | The summarization of benchmarking pipeline. A** Eighteen cellular deconvolution methods, classified based on their data requirements and computational techniques, were evaluated with 50 simulated and real-world spatial transcriptomics datasets. **B** Datasets from six spatial transcriptomics technologies were used for benchmarking, and the scatter plot shows the resolution of each spot and number of spots and genes in each technology. **C** The benchmarking results were measured according to the accuracy, robustness, and usability of the methods.

The performance of each method varied with the data resources used, but some methods performed steadily great accuracy with both simulated and real-world datasets, e.g., Cell2location and DestVI (Fig. 2). Detailed quantifications of JSD, RMSE, and PCC are also provided (Supplementary Figs. 1, 5, 9–11). Using simulated data, most of the methods performed well with MERFISH, but only CARD, DestVI, and SpatialDWLS were high-performing methods with seqFISH+, indicating that they worked well with a low number of spots. When the number of spots was higher (i.e., with MERFISH and Slide-seqV2), Cell2location, SpatialDecon, and Tangram were most capable of performing deconvolution with large views of tissues. In addition, SpatialDWLS performed well with simulated datasets but poorly with all real-world datasets. The whole cell-type proportions and their ground truth for all six cell types in seqFISH+ and 12 samples in MERFISH were visualized among all methods (Fig. 3A, B and Supplementary Figs. 2–4, 11–13). To explore the performances associated with these six cell types, spider plots were produced to show the RMSE of the 18 methods for each cell type (Fig. 3C). Excitatory neurons had the highest cell-type abundance in seqFISH+, and this dataset had the highest RMSE among all methods. The same trend was observed for inhibitory neurons in MERFISH. With the sequencing-based datasets, the PCC of ST did not show a strong relationship and distinct spatial patterns among all methods. In the ST dataset, the lowest resolution of the spots of all existing spatial transcriptomics technologies and a highly heterogenous tissue sample of pancreatic cancer led to the disappearance of

the spatial patterns of individual cell types and blurred the relationships between cell types and selected marker genes. Nevertheless, some of the methods still achieved relatively high PCCs, e.g., CARD, Cell2location, SpatialDecon, and stereoscope. With the Visium dataset, most methods performed well with three paired cell types and marker genes. Spatial patterns were unclear, which was related to the choice of cell types rather than a heterogeneity issue in the datasets. With the Slide-seqV2 and stereo-seq datasets, the spatial patterns were distinct and most of the methods achieved relatively high PCCs, especially Cell2location, STdeconvolve, and RCTD, which were the top-three performing methods. Among the top-three least-performing methods, DestVI tended to output the average cell-type proportions, whereas SpiceMix and SpatialDWLS generated the mapping with much noise which means that they could not distinguish the cell type patterns well in the real-world datasets.

## Robustness

To evaluate the robustness of the 18 tested methods, we designed several experiments with different conditions as follows: (1) 10,000, 6000, or 3000 genes were randomly chosen in the seqFISH+ dataset and 26,365, 18,000, and 9000 in zebrafish embryo dataset by stereo-seq; (2) three resolutions were simulated with 12 MERFISH datasets and zebrafish embryo dataset using binning sizes of 20, 50, and 100 μm and 5, 10, and 15 μm, respectively, which were also used to test performance with different numbers of spots; (3) 17 original cells types

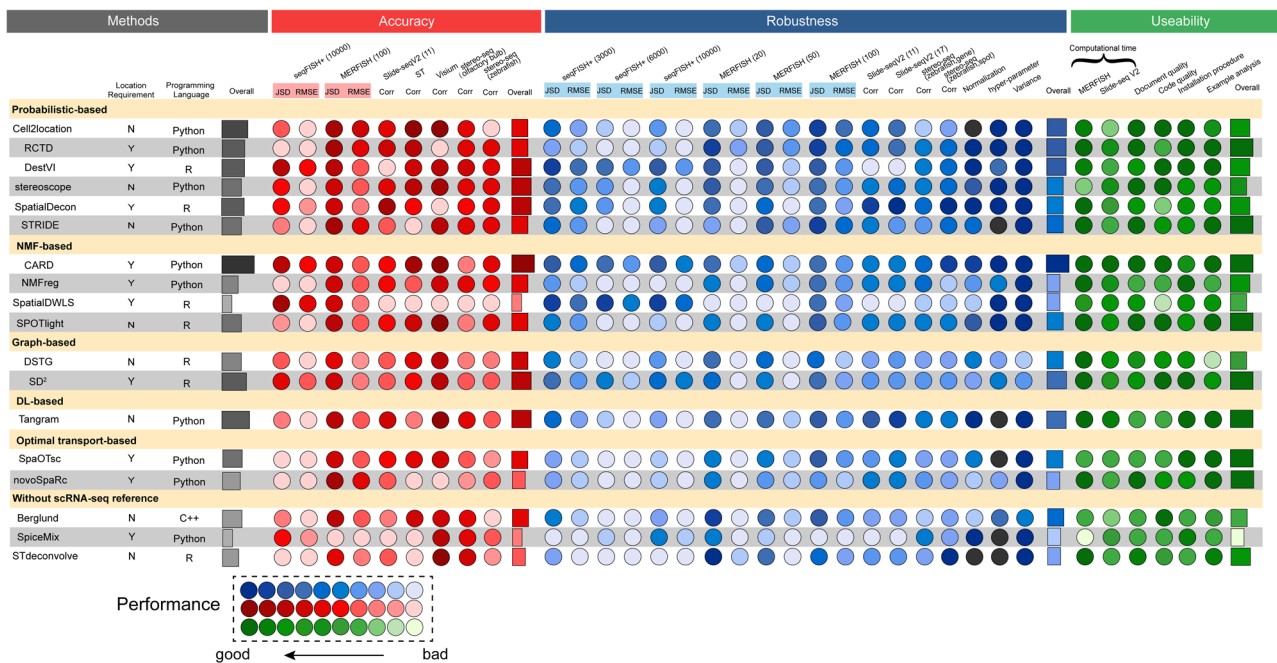

**Fig. 2 | The summary table of the performance of all methods.** We visualized their performance in terms of accuracy (red), robustness (blue), and usability (green), and we listed the requirement of spatial location, programming language, and the overall performance (gray) for each method. For all colored dots, a darker shade represents better performance. The black dots shown in normalization and hyperparameter of robustness meant that the methods required the raw count as input spatial transcriptomics data only or do not have hyperparameters to regulate. Source data are provided as a Source Data file.

and 11 integrated cell types were tested in Slide-seqV2 datasets; (4) two kinds of normalization methods in the Visium dataset to test the effect of normalization of input spatial transcriptomics data; (5) three different values for each chosen hyperparameters in Visium and Slide-seq V2 datasets among five pairs of cell types and their marker genes; and (6) the stability of the performance was assessed by repeating the experiments three times with the seqFISH+ dataset using 10,000 genes per spot.

We found that the general JSD and RMSE of almost all methods were barely changed with the seqFISH+ datasets; the exception was SpiceMix, which performed surprisingly weakly with 3000 genes (Supplementary Fig. 5 and Supplementary Dataset 1). In the tests of different spot numbers and resolutions with MERFISH datasets, the performance of all methods worsened with an increasing number of spots, and this tendency was most evident with SpiceMix and STRIDE (Supplementary Fig. 9 and Supplementary Dataset 1). We visualized the JSD and RMSE using all 12 MERFISH samples with three resolutions and all methods, and the patterns mentioned above were more distinct, although Tangram, stereoscope, and DestVI performed steadily with all 36 datasets (Supplementary Fig. 10). However, on these datasets, DSTG and SpiceMix did not perform well. In particular, DSTG could not identify the clear patterns of different cell types from visualization results and SpiceMix did not perform well possibly due to the misalignment of predicted topics and biological cell types. Using the Slide-seqV2 datasets, we chose two paired sub-cell types from the 17 original cell types and combined each pair as new cell types with biological senses (see "Methods"). For example, two sub-cell types, CA1 and CA3, are the main subfields of the hippocampus proper with their own spatial pattern at the start and end of the neural circuit[31,32]. We combined these two sub-cell types under the name "Cornu Ammonis" (CA), which was the former name of the hippocampus. The spatial patterns from the sub-cell types (CA1 and CA3) and integrated cell types were located from the results of 17 and 11 cell types in Slide-seqV2 (Supplementary Figs. 14 and 15). Based on the visualization and PCCs of the Slide-seqV2 datasets, we found that SpatialDecon, Tangram, and RCTD

had the capability to handle datasets with fine-grained sub-cell types (Supplementary Fig. 11). For the zebrafish embryo dataset by stereo-seq, SpatialDecon, Tangram and CARD showed great performance in terms of both different spot numbers and different gene numbers among three different kinds of cell types (Supplementary Figs. 17 and 18). The effects of normalization methods to the performance were also evaluated on the Visium dataset by comparing the results from input spatial transcriptomics data as the raw count, and two normalization functions: lognorm and sctransform[36]. These two normalization functions were commonly used methods in the analysis of scRNA-seq data. The results showed that many methods (e.g., DestVI, SpatialDecon and SPOTlight) performed better using raw data than using normalized data by lognorm (Supplementary Fig. 19), mainly because there were default normalization procedures in these methods. Thus, lognorm would repeatedly normalize the data which resulted in worse performance than directly inputting the raw data. On the other hand, some of methods (e.g., SpaOTsc and Tangram) did not have any default normalization procedure in their pipelines which caused better performance when being normalized by lognorm. But all the methods performed worse with the sctransform normalization (Supplementary Fig. 19). In this evaluation, Cell2location and STdeconvolve were not included because they required to use the raw count as input data. With respect to the effect of hyperparameters, three different values of each hyperparameter for individual methods were chosen to calculate the variance of PCCs (Supplementary Table 5). This experiment was conducted on Visium and Slide-seq V2 datasets over five pairs of cell types and their marker genes. The results reflected that most of the methods were stable enough among different hyperparameters except DSTG whose variances were over 0.01 (Supplementary Fig. 20). To test the stability of all methods, we repeated the experiments three times with seqFISH+ and 10,000 genes per spot and found that 13 methods showed a steady performance with three identical results. Of the remaining five methods, SD[2] and DSTG exhibited high variance in the JSD and RMSE, which was related to their strategies of using synthesized pseudospots (Supplementary Fig. 21).

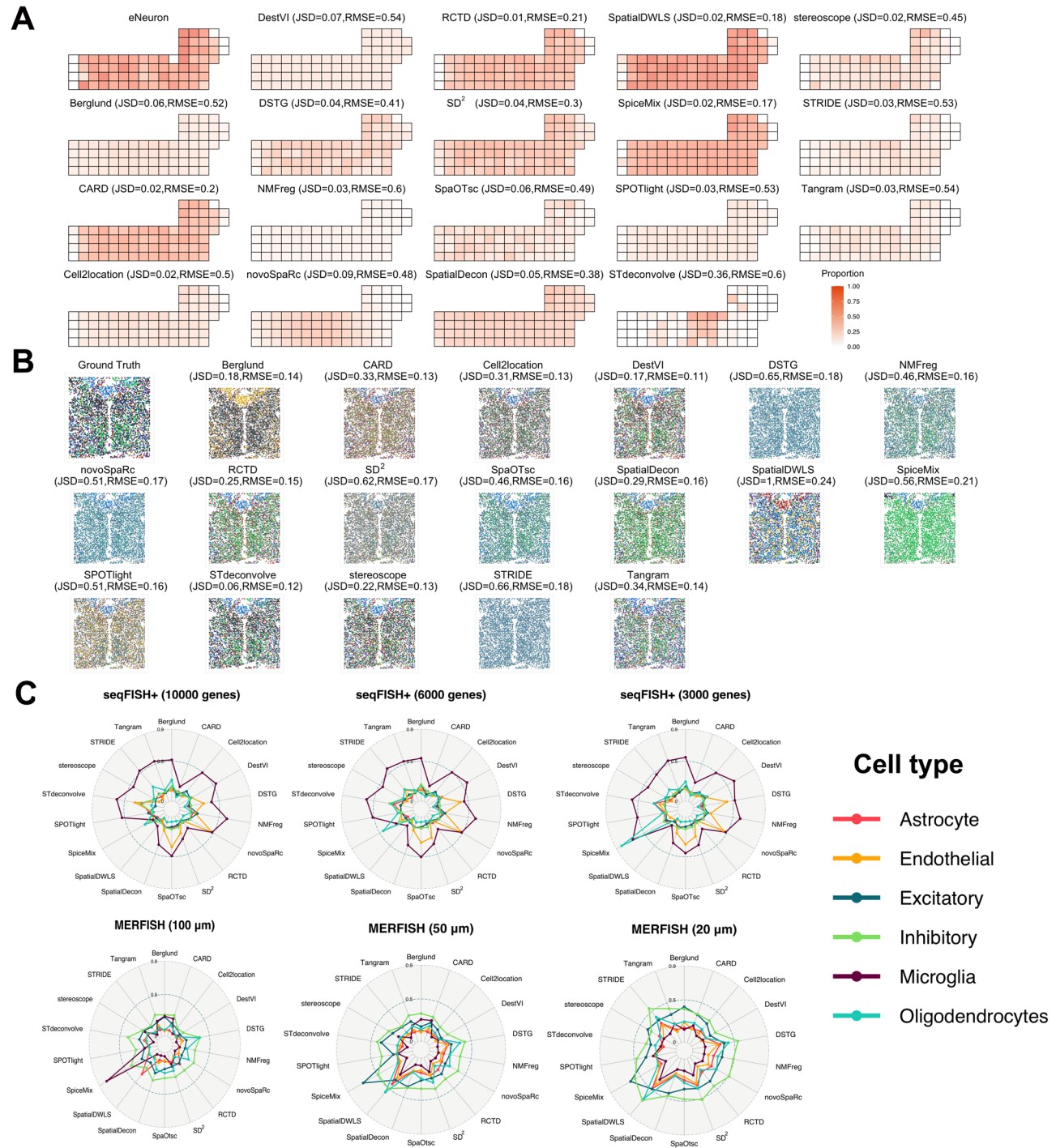

**Fig. 3 | The performance of all methods for simulated datasets. A** Visualization of the ground truth and predicted the proportions of excitatory neurons for 18 methods with the seqFISH+ datasets and 10,000 genes per spot. **B** Visualization of the ground truth and predicted results of deconvolution for all methods with MERFISH datasets (100 μm resolution per spot). The six cell types are represented by the six different colors shown in (**C**). **C** Spider plots showing the RMSE of the deconvolution results for the 18 methods among 6 cell types from the MERFISH (100, 50, and 20 μm resolution per spot) and seqFISH+ (10,000, 6000, and 3000 genes per spot) datasets. Source data are provided as a Source Data file.

Generally, CARD, Cell2location, Tangram, and SD² were the most robust methods according to their performance with different resolutions, number of genes, number of spots, and number of cell types (Fig. 2).

**Usability**

Besides testing the performance of the methods with different situations and datasets, we also assessed their computational efficiency and user-friendliness, which are important factors to users. To fulfill the main concerns of users, we recorded the running time with three different spot numbers in the MERFISH datasets and stereo-seq dataset, and scored several aspects of the tutorials of all methods, including document quality, code quality, installation procedure, compatibility for operating system and example analysis.

According to running time (Supplementary Tables 3 and 4), NMFreg, STRIDE, and Tangram were the most efficient methods. As the

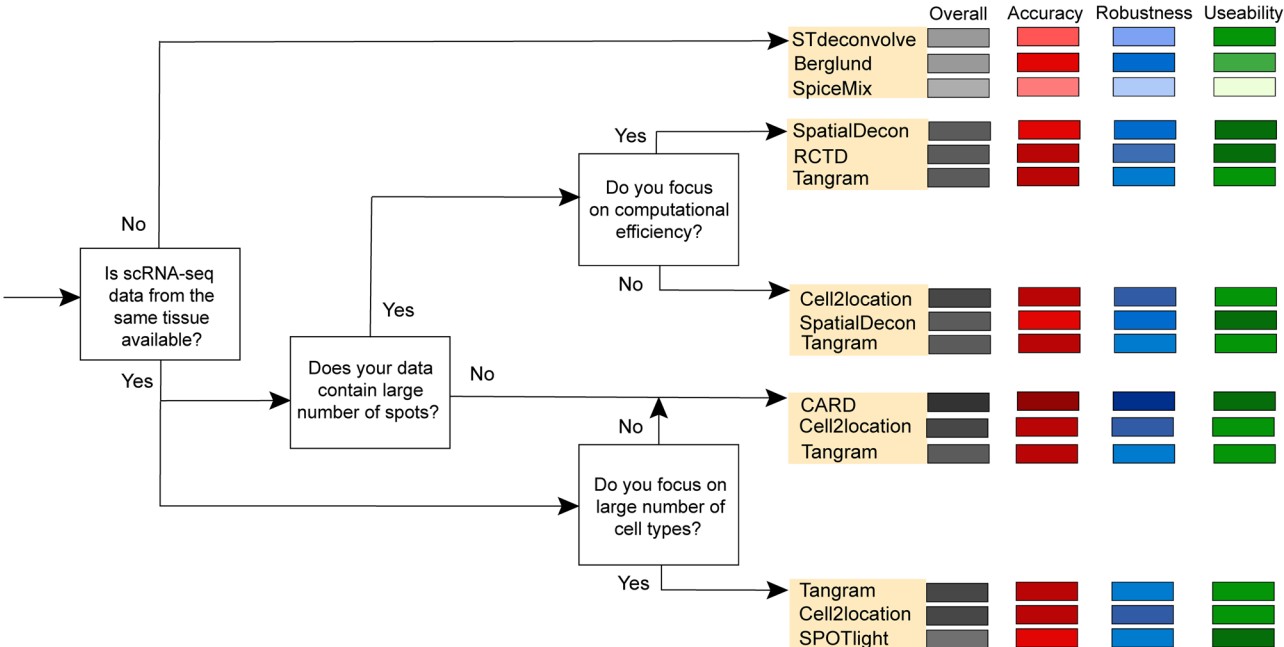

**Fig. 4 | Scenario-specific decision-tree-style guidelines for users.** Four common scenarios are included, and three methods are recommended for each branch.

default hyperparameter is set when using all methods, hyperparameter selection has a substantial effect on method efficiency. In terms of the quality of tutorials and code, most methods satisfied the basic requirements of users. In particular, CARD, Cell2location, RCTD, and DestVI were highly user-friendly with helpful tutorials and readable code that were easy for users to implement.

## Guidelines

Considering the performance of each method and the features described in their related publications, we provided scenario-specific recommendations and guidelines for the methods according to four crucial scenarios (Fig. 4). Because users usually pay less attention to the computational techniques of methods, all scenarios were related to the characteristics of data and computational efficiency. For example, the first scenario was the absence of scRNA-seq reference data from the same tissue, which could be considered a general scenario for users with several possible concerns: (1) the lack of scRNA-seq data from some markedly heterogeneous tissue sample (e.g., cancerous tissue); (2) missing or inconsistent cell types annotated between scRNA-seq and spatial transcriptomics data; and (3) the platform effects of scRNA-seq and spatial transcriptomics data. For each branch of the decision tree, there were three recommended methods. Tangram and Cell2location succeeded in the most situations with the best performance.

Several additional features were described for some methods in their related publications, and these were assessed for our guidelines for method selection. SpatialDecon and RCTD are claimed to correct the variance in gene expression profiles, which resolves the platform effects between scRNA-seq and spatial transcriptomics data[11,14]. Cell2location and SpiceMix can identify potential fine-grained sub-cell types, and CARD can impute cell-type compositions to construct a refined spatial tissue map[9,10,22]. Most methods utilize discrete cell types, although DestVI has the advantage of identifying the continuous variation within the same cell type, which is useful for studying the same tissue section under different conditions[13].

## Discussion

Here, we presented a comprehensive benchmarking study and guidelines for 18 existing cellular deconvolution methods used in

spatial transcriptomics. We evaluated these methods using 50 datasets with multiple metrics in terms of accuracy, robustness, and usability. The datasets we used included simulated datasets binned by single-cell resolution datasets (e.g., seqFISH+ and MERFISH) which could be used for quantitative evaluation as the ground-truth is known for such datasets, and real-world datasets generated by sequencing-based technologies (e.g., Slide-seq V2 and stereo-seq) which could be used to mimic the real-world scenarios for cellular deconvolution tasks. Considering the performance and additional features of the methods, decision-tree-based scenario-specific recommendations and guidelines for method selection were proposed for users. We found that the performance of the 18 methods varied among multiple spatial transcriptomics technologies with different experimental conditions. Nevertheless, each method category contained at least one high-performing method. In general, CARD, Cell2location, Tangram, and RCTD were the best performing methods. Compared with the existing benchmarking studies[8,9], our study included most number of existing methods. More importantly, we provided a full-scale summarization of the performance of all the methods and characterized it as a clear guideline including the solid recommendation of the methods and demonstration of their additional features which would give readers an overall understanding of deconvolution in spatial transcriptomics data analyses.

Following our assessment, we raise two general but crucial limitations that await solutions. First, the platform effect causes two problems as follows: (1) systematic variation in gene expression profiles between scRNA-seq and spatial transcriptomics data; owing to differences in technology-dependent library preparation and sequencing platforms, discrepancies in the detected mRNAs from the same tissue section are inevitable, especially in heterogeneous cancer tissue sections, and (2) variation between two modalities affects the mismatch of cell types between scRNA-seq and spatial transcriptomics data. The prior assumption of integrating these two modalities is to share the same cell types between scRNA-seq and spatial transcriptomics data. Even though RCTD solves the platform effect issue via a normalization strategy among all cell types, cell-type-specific platform effects warrant further exploration[12]. Second, the high dropout rate of spatial transcriptomics is a traditional issue in scRNA-seq. Small libraries lead to deficient mRNA detection; thus, marker

genes for rare cell types become undetectable[16]. The biological pipeline of spatial transcriptomics technologies should be improved further, even though this situation has been considered in SD[2] and an imputation method has been proposed[21,37].

We also present possible future directions of the field to shed light on the development in the field. (1) Multimodal learning will likely become a hotspot in the development of cellular deconvolution methods used in spatial transcriptomics and its applications. For instance, bioinformaticians could use histological tissue images with image intensity levels that could improve our understanding of spatial transcriptomics. (2) Three-dimensional deconvolution and mapping of tissues will provide more novel biological insights than are currently provided by two-dimensional deconvolution. More spatial transcriptomics datasets with consecutive tissue slices are emerging, and the spatial context of interslices will provide more patterns that assist the deconvolution process[11]. (3) Through the development of spatial transcriptomics technologies, the resolution of spots becomes higher, up to the subcellular resolution by recent technologies[34]. Although the rapid progress of spatial transcriptomics technologies is exciting, the marginal benefits of higher resolution are outweighed by the booming higher dropout issue which has more risks of losing some valuable information. In the future, under current sub-resolution technologies, the spatial transcriptomics technology for the single biological cells should be more essential to develop. (4) The cellular deconvolution of spatial transcriptomics will not only help biologists study the structure of tissues but also become associated with artificial intelligence–assisted computational pathology and the healthcare system.

## Methods

### Benchmark metrics

We first assume that there are $J$ genes per spot and $I$ captured spots in the whole spatial transcriptomics data. $X_{ij}$ represents the expression value of gene $j$ in the $i$th spot. $T_{ik}$ and $P_{ik}$ represent the true and predicted proportion of cell type $k$, respectively, in the $i$th spot through the number of total cell types $K$. To evaluate the performance of the tested methods comprehensively, the main benchmark metrics used were RMSE, JSD, and PCC. The definitions of these metrics, as used in our benchmarking pipeline, are provided below.

1. RMSE was calculated between $T_{ik}$ and $P_{ik}$ of per cell type, normalize them by the sum of calculated proportions among all spots $S_k$, and then average them as the final RMSE as the following equation:

$$\text{RMSE} = \sqrt{\frac{1}{K}\sum_{k=1}^{K}\frac{1}{S_k}\sum_{i=1}^{I}\left(P_{ik}-T_{ik}\right)^2} \quad (1)$$

2. JSD was calculated as a score between $T_k$ and $P_k$ per cell type in all spots. The conception of Kullback–Leibler divergence (KL) is used for calculating JSD. $Q(P_k)$ and $Q(T_k)$ represent algorithm-predicted and true distribution of cell type $k$. We average them as the final JSD as the following equation:

$$\text{JSD} = \frac{1}{2}\text{KL}\left(Q(T_k)\parallel\frac{Q(P_k)+Q(T_k)}{2}\right)+\frac{1}{2}\text{KL}\left(Q(P_k)\parallel\frac{Q(P_k)+Q(T_k)}{2}\right) \quad (2)$$

$$\text{KL}(Q(P_k)\parallel Q(T_k)) = \sum Q(P_k)\ln\frac{Q(P_k)}{Q(T_k)} \quad (3)$$

3. Because ground truth does not exist in sequencing-based spatial transcriptomics data, we calculated the PCC between the predicted proportion of specific cell type $P_k$ and the expression profile of its marker gene $E_g$ using the following equation:

$$\text{PCC}\left(P_k, E_g\right) = \frac{\mathbb{E}\left[P_k E_g\right]-\mathbb{E}\left[P_k\right]\mathbb{E}\left[E_g\right]}{\sqrt{\mathbb{E}\left[P_k{}^2\right]-\left(\mathbb{E}\left[P_k\right]\right)^2}\sqrt{\mathbb{E}\left[E_g{}^2\right]-\left(\mathbb{E}\left[E_g\right]\right)^2}} \quad (4)$$

### Evaluation of methods without a scRNA-seq reference

The methods used without a scRNA-seq reference, also called unsupervised methods, deconvolved low-resolution spots based on the gene expression profile and location of spots from spatial transcriptomics data only. STdeconvolve was inspired by the notion of discovering latent topics in collections of documents, which is a common task in natural language processing, and uses Latent Dirichlet Allocation to infer the proportions of cell types based on gene expression profiles in spatial transcriptomics. Berglund uses Poisson factor analysis and Monte-Carlo Markov Chain sampling to deconvolve spots. SpiceMix uses the locations of spots as an extra input in addition to gene expression profiles and incorporates graph representations of spatial relationships into matrix factorization to deconvolve spots.

In general, unsupervised methods needed user to specify the number of topics which represented the clusters waiting for assigning the names of known cell types. Ideally, the number of topics should be similar to the number of cell types. In our evaluation, we set the number of topics as the true number of cell types manually among all datasets. After the deconvolution, a topic-by-spot matrix was generated, and we multiplied this matrix and inputted a spot-by-gene matrix to obtain the topic-by-gene matrix. We aimed to map the topics to real cell types for further evaluation. First, we summed up the same cell types in an annotated cell-by-gene matrix from the scRNA-seq data as the cell-type-by-gene matrix. For each real cell type from scRNA-seq data, we calculated the PCC between this cell type and all topics, chose the best-paired topic with the highest PCC, and assigned the name of current cell type to chosen topic. After assignment, this chosen topic would be ignored in the future steps. Then, we repeated the aforementioned steps on the next cell type until all cell types were iterated. For now, each topic should be paired with the best suitable cell type without duplication and topic-by-spot matrix could be easily transferred to cell-type-by-spot matrix for evaluating the performance of unsupervised methods further.

### Preprocessing of datasets

Owing to the extremely high resolution (500 nm per spot) and dropout rate of stereo-seq data, it was necessary to integrate subcellular-resolution spots into low-resolution spots by binning them using a $100\times100$ spot square (bin100) with slides of $50\,\mu m$ ($100\times500$ nm) and summing their gene expression profiles. The bin100 stereo-seq data had a similar resolution as that of Visium, and it performed reasonably in deconvolution tasks. For the zebrafish embryo dataset by stereo-seq, it was binned by 5, 10 and 15 μm to test the robustness.

To evaluate the robustness of the methods, we tested their performance in terms of different cell types. We integrated the 17 original cell types into 11 cell types, thereby combining some sub-cell types. We integrated CA1 and CA3 into "Cornu Ammonis"; Neuron.Slc17a6, Neurogenesis, and Cajal_Retzius into "Neuron"; Endothelial_Stalk and Endothelial_Tip into "Endothelial"; and Oligodendrocyte and Polydendrocyte into "Oligo_Poly"[12].

For marker gene selection in all datasets, most biological marker genes were chosen from publications. Specifically, we chose the top-five highly variable genes (calculating the fold-change of each gene) for each specific cell type from Slide-seqV2 datasets as the marker genes.

## Construction of a summary table

We constructed a summary table to show the performances of the methods (Fig. 2). Because JSD, RMSE, and running time showed better performance with lower values, we normalized the value $x$ in each column according to $\text{minmax}(\max(x_{col}) - x)$, where $x_{col}$ represents the vector of the column. For the other metrics, we normalized according to minmax directly. Thus, we unified a pattern in which darker dots represent better performance, which is a pattern that users will find easy to identify.

## Implementation of methods

CARD[10]: we used the code of CARD v1.0.0 from https://github.com/YingMa0107/CARD. We set minCountGene to 5 and minCountSpot to 5, which are the default parameter settings.

SPOTlight[19]: we used the code of SPOTlight v0.99.0 from https://github.com/MarcElosua/SPOTlight. We set cl_n to 10 and hvg to 2000.

DSTG[20]: we used the code of DSTG from https://github.com/Su-informatics-lab/DSTG. We set learning_rate to 0.01 and epoch to 300.

SpatialDWLS[18]: we used the code of SpatialDWLS from https://github.com/RubD/Giotto/, which integrates the SpatialDWLS method. We set min_genes in findMarkers_one_vs_all to 20 and gene_det_in_min_cells and min_det_genes_per_cell in filterGiotto to 5 and 5, respectively.

SD[2 21]: we used the code of SD[2] from https://github.com/leihouyeung/SD2, with the following settings: spot_num = 1000, lower_cellnum = 2, and upper_cellnum = 20.

NMFreg[38]: we used the code of NMFreg from https://github.com/tudaga/NMFreg_tutorial. The NMF function was used with the following parameters: number of components = 30, random_state = 17, and init = random.

Stereoscope[14]: we used the code of stereoscope v.03 from https://github.com/almaan/stereoscope. Analysis with stereoscope was conducted on a GPU with the following parameters: number of genes = 5000, st epochs = 75,000, st batch size = 1000, sc epochs = 75,000, sc batch size = 1000, and learning rate = 0.01.

Tangram[22]: we used the code of Tangram v1.0.3 from https://github.com/broadinstitute/Tangram. The mapping of cells to space was conducted with the function tg.map_cell_to_space with mode = clusters.

Cell2location[11]: we used the code of Cell2location v0.1 from https://github.com/BayraktarLab/cell2location. The settings max_epochs = 4000, batch_size = None, and train_size = 1 were used.

STdeconvolve[25]: we used the code of STdeconvolve 1.0.0 from https://github.com/JEFworks-Lab/STdeconvolve. We used the default settings, except that the number of factors was set correctly according to each dataset.

Berglund[23]: we used the code of Berglund 0.2.0 from https://github.com/SpatialTranscriptomicsResearch/std-poisson. We set the following parameters: –iter (=2000), --feature_alpha arg = 1, --mix_alpha arg (= 0.5), --phi_r_1 arg (= 1), --phi_r_2 arg (= 0.001), --phi_p_1 arg (=2), --phi_p_2 arg (=2), --theta_r_1 arg (=1), --theta_r_2 arg (=1), --theta_p_1 arg (= 0.050000), --theta_p_2 arg (= 0.950000), --spot_1 arg (=10), --spot_2 arg (= 10), --sigma arg (=1), --residual arg (= 100), --bline1 arg (=50), and --bline2 arg (=50) The MCMC inference options were set as follows: --MHiter arg (=100) and --MHtemp arg (= 1).

SpiceMix[24]: we used the code of SpiceMix from https://github.com/ma-compbio/SpiceMIx. We selected the number of factors based on the used dataset and set use_spatial to True.

RCTD[12]: we used the code of RCTD from https://github.com/dmcable/spacexr, which is integrated into a tool called spacexr (2.0.0). Spacexr (RCTD) was run with following the configuration: (1) create.RCTD was used with the parameter CELL_MIN_INSTANCE = 1; (2) run.RCTD was used in the doublet mode.

SpatialDecon[15]: we used the code of SpatialDecon from https://github.com/Nanostring-Biostats/SpatialDecon.git. SpatialDecon was run with the expected background count bg set to 0.01.

STRIDE[16]: we used the code of STRIDE from https://github.com/DongqingSun96/STRIDE. The cell-type-associated topic profiles were obtained using the "STRIDE deconvolve" function. If not specified, STRIDE set the 75% quantile of nCount as the default scaling factor.

DestVI[13]: we used the code of DestVI (scvi-tools 0.16.0) from https://github.com/scverse/scvi-tools. DestVI first required genes with <10 counts to be filtered using the function "sc.pp.filter_genes." To perform deconvolution, the single-cell model was then trained to learn the basis of gene expression with the scRNA-seq data for 300 epochs, whereas the spatial model was trained for 2500 epochs, with a learning rate of $10^{-3}$.

SpaOTsc[26]: we conducted the code of SpaOTsc from https://github.com/zcang/SpaOTsc.

novoSpaRc[27]: we conducted the code of novoSpaRc 0.4.4 from https://github.com/rajewsky-lab/novosparc. Alpha was set as 0.5.

## Computational resource

The workstation used to test all methods had a 2 Intel(R) Xeon(R) CPU E5-2680 v3 @ 2.50 GHz (30,720 KB cache size; 24 cores in total) and 528 GB of memory. The GPUs were two Nvidia Quadro M6000 24 GB (48 GB in total). The operating system used was Ubuntu 18.04.

## Reporting summary

Further information on research design is available in the Nature Portfolio Reporting Summary linked to this article.

## Data availability

A summary of the data is shown in Supplementary Table 1. seqFISH+: scRNA-seq and spatial transcriptomics data were both obtained from https://github.com/CaiGroup/seqFISH-PLUS. MERFISH: scRNA-seq data were obtained from https://github.com/rdong08/spatialDWLS_dataset/tree/main/datasets, and spatial transcriptomics data were obtained from https://datadryad.org/stash/dataset/doi:10.5061/dryad.8t8s248/. ST: scRNA-seq and spatial transcriptomics data were both obtained from GSE111672. Visium: scRNA-seq data and spatial transcriptomics data were obtained from https://github.com/BayraktarLab/cell2location. Slide-seqV2: scRNA-seq and spatial transcriptomics data were both obtained from https://github.com/dmcable/spacexr. Stereo-seq (olfactory bulb): scRNA-seq data were obtained from GSE71585, and spatial transcriptomics data were obtained from https://db.cngb.org/stomics/mosta/. Stereo-seq (zebrafish embryo): the scRNA-seq data and spatial transcriptomics data are from https://db.cngb.org/stomics/datasets/STDS0000057. All the mentioned datasets were also integrated and uploaded to public repository[39]. All other relevant data supporting the key findings of this study are available within the article and its Supplementary Information files or from the corresponding author upon reasonable request. No data were excluded from the analyses; the experiments were not randomized; the Investigators were not blinded to allocation during experiments and outcome assessment. Source data are provided with this paper.

## Code availability

The code is available at https://github.com/leihouyeung/STdeconv_benchmark[39].

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

## Acknowledgements

This publication is based upon work supported by the King Abdullah University of Science and Technology (KAUST) Office of Research Administration (ORA) under Award Nos. FCC/1/1976-44-01, FCC/1/1976-45-01, URF/1/4663-01-01, REI/1/5202-01-01, REI/1/5234-01-01, REI/1/4940-01-01, and RGC/3/4816-01-01. We thank Hanmin Li for helping running some experiments.

## Author contributions

X.G. and H.L. conceived and initiated this study. X.L., B.Z., R.Z. and Y.W. prepared all spatial transcriptomics datasets and scRNA-seq datasets. H.L., J.Z., Z.L. and S.C. conducted all the experiments of all methods under accuracy, robustness and usability. H.L. and J.Z. outputted the figure and tables. H.L. wrote the manuscript under supervision of X.G. S.S. polished the writing of manuscript. All authors are involved in discussion and finalization of the manuscript.

## Competing interests

The authors declare no competing interests.
