## [Peer Review File · Nature Communications]

A comprehensive benchmarking with practical guidelines for cellular deconvolution of spatial transcriptomicsReviewer #1 (Remarks to the Author):

In this paper, the authors benchmarked 16 existing methods and proposed clear guidelines for solving the deconvolution problem in spatial transcriptomics (ST) data analysis, which is one of the fastest developing biotechnologies. In addition to the traditional "omic" technologies, ST could add another dimension, spatial location, to help characterize the transcriptional and spatial pattern of the tissue of interest. It was named Method of the year 2020 by Nature Methods and various computational methods have been developed to tackle different key problems in ST. Therefore, this benchmarking study is timely needed and will be very useful to readers and users of these tools.

In general, the paper is very well written, and the experiments are carefully designed and comprehensively conducted. In addition to in-depth comparison and analysis of the results, the authors gave clear guidelines, suggestions, and recommendations on the usage of different methods; it is expected to greatly benefit the community. In particular, they compared all the existing methods in three aspects, accuracy, robustness, and usability across a large number of ST datasets that cover different sequencing technologies, resolutions, spot numbers, and gene numbers. Based on the comparative results, they further proposed scenario-specific guidelines with four general scenarios so that researchers can easily follow.

I have no doubt that this work will make important contributions to the community, but I would like the authors to further improve their work according to the following comments:

1. In Figure 1C, "predicted cell type" should be changed to "predicted proportion".
2. In addition to presenting the analytical results on real-world datasets, please add explanations and discussions on why some methods perform well or poorly on different datasets.
3. The caption of Figure 4 is not clear, please rewrite it.
4. The order of the top-three recommended methods should follow the same order as their reported performance in Figure 4.
5. In the benchmark metrics, every formula needs to be numbered at right.
6. In the caption of Figure 2B, "Visualization of the ground truth and deconvolution results" should be changed to "Visualization of the ground truth and predicted results of deconvolution".
7. In the evaluation of robustness, there were some methods which could not reach good performance, especially on the MERFISH dataset. What is the reason for such bad performance?
8. The authors proposed three future directions in ST. Since the development of ST techniques is towards the higher resolution, i.e., to single-cell or even sub-cell resolution, more discussions about the perspective on the influence of resolution on methodology development should be added.

Reviewer #2 (Remarks to the Author):

Spatial transcriptomics could profile cells' expression as well as their spatial contexts. However, currently, the whole transcriptome-based spatial transcriptomics data suffers from low resolution as each spatial pixel contains more than one cells. Many deconvolution algorithms exist, and the field is in need of a study for evaluation of different spatial transcriptomics data deconvolution algorithms. This study comprehensively evaluated 16 existing spatial deconvolution methods with a large number of real world and simulated datasets in terms of methods' accuracy, robustness, and usability. Overall, the significance of the work is undoubted, and the presentation of the work is of high quality. There are some comments listed below warranting further considerations.

Major comments:

In Methods/Benchmarking Metrics section, RMSE, JSD and PCC were defined for true and predicted proportions. I have a few related questions.

For RMSE, its definition is fine. However, the square root of average sum-of-square error over all cell types may be problematic, as some cell types with high abundance may dominate the metric; in other words, it may favor an algorithm that predicts well cell types with high proportion.

For JSD, its definition seems to be problematic. KL-divergence is often used to measure the difference between two distributions; however, in JSD, it is not clear what is $(Q(P_k) + Q(T_k))/2$?

It is certainly not a distribution anymore. I wonder why don't the authors use the average of $D_{KL}(Q(P_k)||Q(T_k))$ and $D_{KL}(Q(T_k)||Q(P_k))$. In addition, "Q represents the probability distribution of a vector" doesn't make statistical sense, as the distribution of a vector will be a multi-variate distribution. I assume that the authors wanted to say "Q(P_k) and Q(T_k) represent the algorithm-predicted and true distribution of cell type k". Here, the use of average over all cell types, again, would be problematic under cell type bias similar to RMSE. For all metrics, aside from the average metric for all cell types, it would be nice to also have the individual cell type metric, i.e., RMSE/JSD/PCC for cell typ1, ..., cell type K. Such information would be useful for users to look for the methods they need.

Minor comments:

The authors claimed that there are 44 datasets. It would help if they could provide a table to detail these datasets, aside from supp table 1.

Supp Figure 16, it should be "three cell types" instead of four cell types.

Supp Figure 17, some methods have no variance, I wonder why would that be. Across the three simulations, is the dataset the same?

For unsupervised methods, it is hard to annotate the cell types. According to the authors, using annotated single cell data, a "topic" is annotated to be the one with the highest correlation. This seems to be a little arbitrary, and could sometimes be problematic. Did two topics ever point to the same cell type?

Reviewer #3 (Remarks to the Author):

This study comprehensively benchmarked 16 existing methods for cell type deconvolution using 44 real-world and simulated spatial transcriptomic (ST) datasets, by evaluating the accuracy, robustness, and usability of each method. The authors compared these methods with different resolutions, ST technologies, spot numbers, and gene numbers; and provided scenario-specific recommendations and guidelines for users to choose a most proper method according to four crucial scenarios. It is important to perform benchmark studies over ST analysis tools due to the rapid development of the field and meanwhile not an easy task to benchmark such a diverse of methods. However, the manuscript at the current stage has a number of issues that need to be fully addressed before considering for publication.

1. There are two benchmark papers published earlier this year: the first paper systematically evaluated 12 cellular deconvolution methods in terms of accuracy, robustness and time complexity, using 32 simulated datasets and other sequencing-based ST datasets (PMID: 35577954, Nat. Methods); and the other paper assessed the performance of 10 methods for ST deconvolution, with either ideal reference or non-ideal reference (PMID: 35753702, Briefings in Bioinformatics). The authors are apparently aware of the first paper, which was cited in the reference. Nevertheless, these two published studies significantly reduce the novelty of this study, and thereby the authors need to clarify the similarities and especially the uniqueness and new discoveries of this study compare with these two published papers. The value and novelty of this study should also be claimed clearly in the Results or Discussion of the manuscript.

2. The author claimed that "Real-world sequencing-based spatial transcriptomics data cannot be superseded by simulated data; thus, researchers cannot choose an optimal method by generalizing results from simulated data to real-world sequencing-based scenarios". I agree that this is an important point for benchmarking. However, in this manuscript, the authors used only 4 real-world sequencing-based ST data for accuracy evaluation. Such a small datasets cannot support the conclusion. Therefore, the authors need to collect more real-world datasets for benchmarking. Besides, although the authors claimed that "44 real-world and simulated datasets" were used for benchmarking, in fact, only six image-based and sequencing-based ST datasets were used for accuracy evaluation. The author needs to used more datasets, for instance, include 45 real datasets or 32 simulated datasets used in this study (PMID: 35577954, Nat. Methods).

3. Related to point 2, when evaluating the robustness of the 16 tested methods, the authors simply used simulated datasets synthesized from image-based data and Slide-seqV2 data. What is the role of real-world sequencing-based ST data when evaluating the robustness? And how robust

are these deconvolution methods on real-world sequencing-based ST data?

4. With sequencing-based ST data, the author calculated the PCC between the predicted proportion of specific cell type and the expression profile of its corresponding marker genes to quantify the prediction accuracy. I am a little concerned about this hypothesis, although it does make sense that the marker gene expression and deconvolution results have a similar pattern in space, they are not supposed to be perfectly and directly correlated which can be used as gold-standard for benchmarking. Additionally, the normalization methods used to process ST data from low-resolution sequencing-based technologies such as 10X Visium, can affect the values of PCC. I am wondering if similar results will still be obtained using different normalization method? And if the benchmarking results are stable with 10, 20 or more marker genes to calculate the PCC?

5. It is interesting that the authors evaluated the impact of the number of spot, number of gene, and resolutions on these integration methods. Are all of these methods expected to perform similarly on different datasets? Again the authors need to include more dataset in each dimension of the comparison to reduce any bias raised by datasets and thereby achieve more convincing conclusions.

6. In this benchmarking, the authors run each method with a single choice of parameters, but it seems reasonable that each method will reach the best performance under certain parameters due to the data property or technologies that the data is generated. For example, the "batch-size" parameter in destVI needs to be adopted to work better with the datasets that contain less spots. The authors need to compare the performance of the methods where optimized parameters of each method were applied.

7. There are some spots which contains more cells/cell-types and some contains fewer number of cells/cell-types (Figure 3C), I am wondering how does the number of cells and cell-types affect the performance of these integration methods? Is it possible that some cell subtypes either with greater or less proportion in ST spots, can reach higher prediction accuracy by all integration methods?

8. As a complement to evaluation of methods on real data, the authors might consider an evaluation on simulated data where a ground truth is known. Although simulated data obviously cannot match all of the features of real data, the ground truth represents a gold-standard that is particularly important to draw convincing conclusions in benchmarking studies.

9. Perhaps my own lack of knowledge, it is not clear to me why the authors excluded novoSpaRc and SpaOTsc in the benchmarking. These two methods are capable of allocating cells to spatial positions and therefore have wide applications in general.

10. The evaluation of the usability (Figure 2, which should be a table instead) is insufficient. Why not all the datasets were assessed to compare the usability? Will the number of cell types and shared genes effect the usability? Could the authors clearly define the user-friendliness for each method? Why some are more "friendly" than the others?

11. The readability of some figures, such as Fig3A, B and Supplementary Figure 6,7,8, are terrible. No corresponding metrics or statistics panels were included on the figures, making readers hard to follow.

In general I found this study important and interesting, however, a big chunk of novelty was taken by the two published papers earlier this year, and the manuscript at the current stage is still premature where a lot of technical details need to be further implemented.

We are very grateful to the three reviewers for their thoughtful and thorough comments, which definitely helped us improve our paper greatly. We have revised the paper following all of their comments. Below please find the point-by-point response to all the reviewers' comments.

Reviewer #1 (Remarks to the Author):

In this paper, the authors benchmarked 16 existing methods and proposed clear guidelines for solving the deconvolution problem in spatial transcriptomics (ST) data analysis, which is one of the fastest developing biotechnologies. In addition to the traditional “omic” technologies, ST could add another dimension, spatial location, to help characterize the transcriptional and spatial pattern of the tissue of interest. It was named Method of the year 2020 by Nature Methods and various computational methods have been developed to tackle different key problems in ST. Therefore, this benchmarking study is timely needed and will be very useful to readers and users of these tools.

In general, the paper is very well written, and the experiments are carefully designed and comprehensively conducted. In addition to in-depth comparison and analysis of the results, the authors gave clear guidelines, suggestions, and recommendations on the usage of different methods; it is expected to greatly benefit the community. In particular, they compared all the existing methods in three aspects, accuracy, robustness, and usability across a large number of ST datasets that cover different sequencing technologies, resolutions, spot numbers, and gene numbers. Based on the comparative results, they further proposed scenario-specific guidelines with four general scenarios so that researchers can easily follow.

First of all, we would like to thank you very much for acknowledging our contributions and for your very helpful comments. We have followed all your suggestions and comments to revise our manuscript, and we believe that it has greatly improved the quality of our paper. Below are the detailed responses to each of your comments.

I have no doubt that this work will make important contributions to the community, but I would like the authors to further improve their work according to the following comments:

1. In Figure 1C, “predicted cell type” should be changed to “predicted proportion”.

Thanks for pointing out this issue. We have revised it in the manuscript.

2. In addition to presenting the analytical results on real-world datasets, please add explanations and discussions on why some methods perform well or poorly on different datasets.

Thank you for the suggestion. We have added more explanations and discussions on this point in the Results section.

3. The caption of Figure 4 is not clear, please rewrite it.

Thank you for pointing this out. We have rewritten the caption of Figure 4.

4. The order of the top-three recommended methods should follow the same order as their reported performance in Figure 4.

Thank you for the good catch. We have revised the order of recommended methods in Figure 4 to make them consistent.

5. In the benchmark metrics, every formula needs to be numbered at right.

Thanks for the comment. We have added the number of each formula in the manuscript.

6. In the caption of Figure 2B, “Visualization of the ground truth and deconvolution results” should be changed to “Visualization of the ground truth and predicted results of deconvolution”.

Thanks for the comment. We have revised it in Figure 2.

7. In the evaluation of robustness, there were some methods which could not reach good performance, especially on the MERFISH dataset. What is the reason for such bad performance?

Thank you for the question. We have added possible reasons for the bad performance of some of methods, such as DSTG and SpiceMix.

8. The authors proposed three future directions in ST. Since the development of ST techniques is towards the higher resolution, i.e., to single-cell or even sub-cell resolution, more discussions about the perspective on the influence of resolution on methodology development should be added.

Thank you for the excellent suggestion. We have elaborated more on the perspective on the higher resolution of spatial transcriptomics technologies.

Reviewer #2 (Remarks to the Author):

Spatial transcriptomics could profile cells’ expression as well as their spatial contexts. However, currently, the whole transcriptome-based spatial transcriptomics data suffers from low resolution as each spatial pixel contains more than one cells. Many deconvolution algorithms exist, and the field is in need of a study for evaluation of different spatial transcriptomics data deconvolution algorithms. This study comprehensively evaluated 16 existing spatial deconvolution methods with a large number of real world and simulated datasets in terms of methods’ accuracy, robustness, and usability. Overall, the significance of the work is undoubted, and the presentation of the work is of high quality. There are some comments listed below warranting further considerations.

Thank you very much for your support on our work and for the very helpful comments. We have followed all your suggestions and comments to revise our manuscript, and we believe that it has greatly improved the quality of our paper. Below are the detailed responses to each of your comments.

Major comments:

In Methods/Benchmarking Metrics section, RMSE, JSD and PCC were defined for true and predicted

proportions. I have a few related questions.

1. For RMSE, its definition is fine. However, the square root of average sum-of-square error over all cell types may be problematic, as some cell types with high abundance may dominate the metric; in other words, it may favor an algorithm that predicts well cell types with high proportion.

Thank you very much for pointing out this issue. You are absolutely right: all such average-based metrics have the issue of being sensitive to the high-abundance cell types. Following your suggestion, we now normalize the RMSE for each cell type by dividing by the sum of calculated proportions among all spots S_k which could eliminate the effect of the abundance issue of cell types.

2. For JSD, its definition seems to be problematic. KL-divergence is often used to measure the difference between two distributions; however, in JSD, it is not clear what is $(Q(P_k)+Q(T_k))/2$? It is certainly not a distribution anymore. I wonder why don't the authors use the average of $D_{KL}(Q(P_k)||Q(T_k))$ and $D_{KL}(Q(T_k)||Q(P_k))$. In addition, "Q represents the probability distribution of a vector" doesn't make statistical sense, as the distribution of a vector will be a multi-variate distribution. I assume that the authors wanted to say "Q(P_k) and Q(T_k) represent the algorithm-predicted and true distribution of cell type k". Here, the use of average over all cell types, again, would be problematic under cell type bias similar to RMSE.

Thank you very much for the excellent comment. Firstly, please allow us to explain the reason why we used $(Q(P_k)+Q(T_k))/2$ in the formula. According to the definition of JSD score (https://en.wikipedia.org/wiki/Jensen%E2%80%93Shannon_divergence), $JS(P || Q) = 1/2 * KL(P || M) + 1/2 * KL(Q || M)$ where $M = 1/2 * (P + Q)$. And we just followed this definition and replaced P and Q as the predicted and ground truth cell-type proportions.

Secondly, we have followed your comment to change the "Q represents the probability distribution of a vector" to "Q(P_k) and Q(T_k) represent the algorithm-predicted and true distribution of cell type k". Thank you very much for pointing this out.

Finally, we have simulated and evaluated different abundance of several different cell types and found that the abundance issue would not affect the JSD score which means that the average of JSD score over all cell types is not sensitive to high-abundance cell types whereas the RMSE is.

3. For all metrics, aside from the average metric for all cell types, it would be nice to also have the individual cell type metric, i.e., RMSE/JSD/PCC for cell type1, ..., cell type K. Such information would be useful for users to look for the methods they need.

Thank you for the suggestion. We have shown the radar plots in Figure 3 to show the results of RMSE for simulated datasets through each cell type. We now also supplied the files including both JSD score and RMSE values for individual cell types which could be useful for the readers and users. In the original submission, PCC was already calculated for individual cell types.

Minor comments:

The authors claimed that there are 44 datasets. It would help if they could provide a table to detail these datasets, aside from supp table 1.

Thank you for the comment. Now Table 1 in Supplementary Materials shows the information of each dataset in details.

Supp Figure 16, it should be “three cell types” instead of four cell types.

Thanks for pointing this out. We have revised it.

Supp Figure 17, some methods have no variance, I wonder why would that be. Across the three simulations, is the dataset the same?

Thank you for the question. The reason why some methods have no variance is that there is no randomness through the deconvolution procedures of these methods. On the other side, methods such as DSTG which randomly choose the single cell data to integrate a pseudo-spot causes the randomness in the pipeline.

For unsupervised methods, it is hard to annotate the cell types. According to the authors, using annotated single cell data, a “topic” is annotated to be the one with the highest correlation. This seems to be a little arbitrary, and could sometimes be problematic. Did two topics ever point to the same cell type?

Thank you for the comment. The detailed procedure for pairing cell types and topics is that for each known cell type, we calculated the PCC between this cell type and all topics, chose the best-paired topic with the highest PCC, and assigned the name of current cell type to chosen topic. After assignment, this chosen topic would be ignored in the future steps. Then, we repeated the aforementioned steps on the next cell type until all cell types were iterated. For now, each topic should be paired with the best suitable cell type without duplication which meant that two topics would not point to the same cell type. We have clarified this procedure in the revision.

=====
Reviewer #3 (Remarks to the Author):

This study comprehensively benchmarked 16 existing methods for cell type deconvolution using 44 real-world and simulated spatial transcriptomic (ST) datasets, by evaluating the accuracy, robustness, and usability of each method. The authors compared these methods with different resolutions, ST technologies, spot numbers, and gene numbers; and provided scenario-specific recommendations and guidelines for users to choose a most proper method according to four crucial scenarios. It is important to perform benchmark studies over ST analysis tools due to the rapid development of the field and meanwhile not an easy task to benchmark such a diverse of methods. However, the manuscript at the current stage has a number of issues that need to be fully addressed before considering for publication.

Thank you very much for your very helpful and constructive comments. We have followed all your suggestions and comments to revise our manuscript, and we believe that it has greatly improved the quality of our paper. Below are the detailed responses to each of your comments.

1. There are two benchmark papers published earlier this year: the first paper systematically evaluated 12 cellular deconvolution methods in terms of accuracy, robustness and time complexity, using 32

simulated datasets and other sequencing-based ST datasets (PMID: 35577954, Nat. Methods); and the other paper assessed the performance of 10 methods for ST deconvolution, with either ideal reference or non-ideal reference (PMID: 35753702, Briefings in Bioinformatics). The authors are apparently aware of the first paper, which was cited in the reference. Nevertheless, these two published studies significantly reduce the novelty of this study, and thereby the authors need to clarify the similarities and especially the uniqueness and new discoveries of this study compare with these two published papers. The value and novelty of this study should also be claimed clearly in the Results or Discussion of the manuscript.

Thank you very much for the comment. We have now clarified and emphasized more about the novelty of our study compared with the two benchmarking studies that you kindly pointed out. In summary, compared with these two studies, other than the comprehensiveness of the methods we compared, the most significant novelty should be that we provided a full-scale summarization in Figure 2 for all methods over different datasets and various conditions, which offered readers and users a clear picture of these methods' performance landscape and guidance about which methods to choose and what parameters to set up in practice. Beyond that, we also provided a guideline for readers including the recommendation of methods and their separate features which could give readers an overall understanding of this field and enable them to make the best choice for their own tasks. A scenario-specific decision-tree-based guideline (Figure 4) provides users concrete yet easy-to-follow recommendations under different application scenarios.

2. The author claimed that “Real-world sequencing-based spatial transcriptomics data cannot be superseded by simulated data; thus, researchers cannot choose an optimal method by generalizing results from simulated data to real-world sequencing-based scenarios”. I agree that this is an important point for benchmarking. However, in this manuscript, the authors used only 4 real-world sequencing-based ST data for accuracy evaluation. Such a small datasets cannot support the conclusion. Therefore, the authors need to collect more real-world datasets for benchmarking. Besides, although the authors claimed that “44 real-world and simulated datasets” were used for benchmarking, in fact, only six image-based and sequencing-based ST datasets were used for accuracy evaluation. The author needs to use more datasets, for instance, include 45 real datasets or 32 simulated datasets used in this study (PMID: 35577954, Nat. Methods).

Thank you very much for the constructive comment, which we fully agree. Following your suggestion, we have added more real-world datasets from the zebrafish embryo by the stereo-seq technology. Based on this real-world dataset, we further evaluated all the benchmarked methods under different resolutions of spots (5 μm , 10 μm and 15 μm) and different numbers of genes to obtain a comprehensive comparison of these methods.

3. Related to point 2, when evaluating the robustness of the 16 tested methods, the authors simply used simulated datasets synthesized from image-based data and Slide-seqV2 data. What is the role of real-world sequencing-based ST data when evaluating the robustness? And how robust are these deconvolution methods on real-world sequencing-based ST data?

Thank you for the questions. The role of real-world ST data for the evaluation of robustness is to test the performance under genuine scenarios in different conditions. This kind of evaluation is

indispensable to measure how stable the methods are in extreme cases, especially when the spot number and the gene number are large.

Thus, we simulated the zebrafish embryo dataset as three kinds of spot resolutions: 5 μ m, 10 μ m and 15 μ m and three kinds of gene numbers in ST data: full, 18000 and 9000. To evaluate the performance, we chose three kinds of cell types and paired marker genes to calculate the PCC. The general results are shown in the following figure (Supp. Figure 11). This figure showed the heatmap of PCCs for all methods under different datasets and conditions. In the same datasets, each row represented one chosen cell type. Each dataset was represented by one kind of color as the legend showed. In these six colors, from the top to bottom, they were zebrafish datasets with three resolutions of spots (5 μ m, 10 μ m and 15 μ m) and three gene numbers (full, 18000 and 9000). In each dataset, three kinds of cell types (three rows in each dataset) including Blood Vasculature, Notochord and YSL were used to evaluate the PCCs.

For most methods, the performance on the same cell type did not show much variance. We could also observe that through the decreasing of spot resolution, the performance of most methods increased. For the different numbers of genes, the performance of most methods did not change much.

To visualize the results of robustness for zebrafish datasets, we conducted visualization of three predicted cell types under three different resolutions of spots (the first figure, Supp. Figure 17) and three numbers of genes (the second figure, Supp. Figures 18). For the first figure, we showed the visualization results of all methods on zebrafish datasets with three kinds of resolutions (5 μ m, 10 μ m and 15 μ m) by three kinds of cell types (Blood Vasculature, Notochord and YSL). Each subfigure included: 1) the spatial expressions of marker genes which could be considered as ground truth; and 2) the predicted results of all methods with the names of methods and calculated PCCs. For the second figure, the layout was the same as the first figure. We showed the visualization results of all methods on zebrafish datasets with three kinds of gene numbers (full, 18000 and 9000) by three kinds of cell types (Blood Vasculature, Notochord and YSL).

Blood Vasculature

Notochord

YSL

Blood Vasculature

Notochord

YSL

4. With sequencing-based ST data, the author calculated the PCC between the predicted proportion of specific cell type and the expression profile of its corresponding marker genes to quantify the prediction accuracy. I am a little concerned about this hypothesis, although it does make sense that the marker gene expression and deconvolution results have a similar pattern in space, they are not supposed to be perfectly and directly correlated which can be used as gold-standard for benchmarking. Additionally, the normalization methods used to process ST data from low-resolution sequencing-based technologies such as 10X Visium, can affect the values of PCC. I am wondering if similar results will still be obtained using different normalization method? And if the benchmarking results are stable with 10, 20 or more marker genes to calculated the PCC?

Thank you very much for this excellent comment, which we fully agree. Following the comment, to test the robustness of different normalization methods, we chose 10X Visium datasets with both the raw count and the ones normalized by lognorm and sctransform functions to evaluate the performance of all methods. In this evaluation, we chose three pairs of cell types and their marker genes, and the results

were shown as following heatmap (Supp. Figure 19). In this heatmap, each color at the top bar represented one kind of cell type and we chose three kinds of cell types: Excitatory neurons of claustrum, Excitatory neurons of thalamus and Oligodendrocytes. For each cell type, there were three columns representing the results from raw counts, lognorm normalization and sctransform normalization from left to right.

From this heatmap, we could observe that lognorm normalization does not affect the performance much compared with using the raw count as input data. But the performance is worse by using sctransform normalization (the third column for each kind of cell type). In this evaluation, Cell2location and STdeconvolve were not included because these two methods required to use raw count as the input data.

As for the number of marker genes you kindly pointed out, the following table showed cell types and corresponding marker genes we used from each real-world dataset (Supp. Table 2). We chose the marker genes for individual cell types from the demonstration of papers we referred [1][2][3][4][5] instead of calculation of high-variable-genes (HVGs) from datasets, because using the calculated HVGs from data would result in unfairness when evaluating the PCCs on the same data once again. Instead, choosing marker genes from the demonstration of previous studies could guarantee the reliability of evaluation on real-world datasets. According to the demonstration of the papers [1][2][3][4][5], only 1 to 5 marker genes could be chosen for individual cell types. Thus, we are sorry that we could not choose a greater

number of marker genes as you suggested (10, 20 and more) to evaluate the stability of all methods.

Techniques	Full name of cell type	Abbreviation of cell type in dataset	Marker genes
ST	Centroacinar ductal cells	Ductal.CRISP3.high.centroacinar.like	CLDN2 [1]
	Cancer clone S100A4	Cancer.clone.A	GAPDH [1]
Visium	Oligodendrocytes	Oligo_1, Oligo_2	Mog, Plp1 [2]
	excitatory neurons of thalamus	Ext_Thal_1, Ext_Thal_2	Synpo2, Ptpn3, Slc17a6 [2]
	excitatory neurons of claustrum	Ext_Claupyr	Nr4a2, Synpr [2]
Slide-seqV2 (11 cell types)	Oligodendrocyte & Polydendrocyte	Oligodendrocyte & Polydendrocyte	Bmp4,Gpr17,Neu4,Susd5,Cspg4,Gm21984,Mog,Hapln2,Ermn,Ma [3]
	Cornu Ammonis	CA	Dcn,Fibcd1,Ndst3,Cds1,Mpped1,Hs3st4,Lpl,Ptgs2,Kcnq5,Lsm11 [3]
Slide-seqV2 (17 cell types)	CA1	CA1	Dcn,Fibcd1,Ndst3,Cds1,Mpped1 [3]
	CA3	CA3	Hs3st4,Lpl,Ptgs2,Kcnq5,Lsm11 [3]
	Oligodendrocyte	Oligodendrocyte	Gm21984,Mog,Hapln2,Ermn,Ma [3]
	Polydendrocyte	Polydendrocyte	Bmp4,Gpr17,Neu4,Susd5,Cspg4 [3]
Stereo-seq (olfactory bulb)	excitatory mitral and tufted (M/T) cells	n-15-M/TC-1, n-16-M/TC-2, n-17-M/TC-3	Cdhr1 [4]
	granule cells	n03-GC-1, n07-GC-2, n09-GC-3, n10-GC-4, n11-GC-5, n12-GC-6, n14-GC-7	Slc32a1 [4]
	olfactory sensory neurons (OSNs)	n01-OSNs	Omp [4]
Stereo-seq (zebrafish)	Notochord	Notochord	Cmn, Col2a1a, Ntd5 [5]
	YSL	Epidermis_egfl6..YSL	Apoa1a, Apoa1b [5]
	Blood Vasculature	Blood.Vessel.Endothelial.Cell	Etv2, Flt4 [5]

5. It is interesting that the authors evaluated the impact of the number of spots, number of gene, and resolutions on these integration methods. Are all of these methods expected to perform similarly on different datasets? Again the authors need to include more dataset in each dimension of the comparison to reduce any bias raised by datasets and thereby achieve more convincing conclusions.

Thank you for the suggestion. As we mentioned in addressing your former comments, we used a new stereo-seq dataset of zebrafish embryo with three kinds of spot resolutions (5 μ m, 10 μ m and 15 μ m) and three values of gene numbers (full, 18000 and 9000) to compare all methods comprehensively in each dimension. We did expect that all methods performed similarly through these different conditions which could prove that methods were stable enough. Through the evaluation on zebrafish embryo datasets (the first figure), there were 6 datasets with different conditions represented by 6 different colors as the legend showed and we chose three cell types (Blood Vasculature, Notochord and YSL) for all datasets where each row represented the result of one cell type. We could clearly see the pattern that all methods performed steadily through three kinds of different gene numbers.

In the second figure, we showed the JSD (top) and RMSE (bottom) results of all methods through all methods of all 36 samples in MERFISH datasets under three kinds of resolution (100 μ m, 50 μ m, 20 μ m) and 12 samples. The larger and darker dots represent the higher RMSE or JSD. Through the increase of spot resolution, the performance of all methods had a decreasing tendency which showed a similar pattern to the increasing resolutions (100 μ m, 50 μ m, 20 μ m) on JSD score and RMSE from experiments on MERFISH datasets (Supp. Figure 10).

6. In this benchmarking, the authors run each method with a single choice of parameters, but it seems reasonable that each method will reach the best performance under certain parameters due to the data property or technologies that the data is generated. For example, the “batch-size” parameter in destVI needs to be adopted to work better with the datasets that contain less spots. The authors need to compare the performance of the methods where optimized parameters of each method were applied.

Thank you for the comment. We indeed conducted the parameter sensitivity analysis to find out the effect of hyperparameters on the stability of performance among all methods. For 13 methods, we chose several hyperparameters with three values each to test the variance of the performance on Slide-seq V2 dataset and 10X Visium dataset. For the other 5 methods, there were no hyperparameters that users can set up. The results were shown in the following figure (Supp. Figure 20) and the hyperparameters we chose were shown in the following table (Supp. Table 4). In the first heatmap, we chose three kinds of cell types in 10X Visium dataset (three columns in the blue group) and two kinds of cell types in Slide-seq V2 dataset (two columns in the pink group).

Methods	Hyperparameter	
SPOTlight	cl_n: 5, 10, 15	hvg:1000, 2000, 3000
DSTG	laerning_rate: 0.01, 0.001, 0.1	epoch: 200, 300, 400
SpatialDWLS	min_gene (findMarkers_one_vs_all): 10, 20, 30	
SD²	spot_num: 500, 1000, 2000,	lower_cellnum = 2, 10, 20 and upper_cellnum = 10, 20, 30
NMFreg	number of components: 30, 25, 35	
Stereoscope	number of genes: 4000, 5000, 6000	st & sc epochs: 30000, 40000, 50000
Cell2location	max_epochs: 3000, 4000, 5000	
RCTD	CELL_MIN_INSTANCE: 1, 2, 3	
SpatialDecon	background count bg: 0.01, 0.02, 0.03	

DestVI	epoch for spatial model = 2000, 2500, 3000	learning_rate = 0.01, 0.001, 0.005
CARD	minCountGene: 5, 10, 15	
Berglund	Iter: 1000, 2000, 3000	
novoSpaRc	Alpha: 0.3, 0.5, 0.7	

As we could see in the first figure, most of variances on different cell types and datasets were under 0.01 which meant hyperparameters did not affect performance on most of the methods. The reason why DSTG had the highest variance was that it used the randomly chosen single cells from scRNA-seq dataset to integrate as pseudo-spots which was full of uncertainty throughout the experiment.

7. There are some spots which contains more cells/cell-types and some contains fewer number of cells/cell-types (Figure 3C), I am wondering how does the number of cells and cell-types affect the performance of these integration methods? Is it possible that some cell subtypes either with greater or less proportion in ST spots, can reach higher prediction accuracy by all integration methods?

Thank you for the questions. For the first question, the number of cells included in one spot could be formulated as the resolution of spots. The following three figures showed the effects of resolution of spots and the number of cell types (the first two figures: MERFISH datasets, the third figure: zebrafish embryo dataset).

In the first figure, the spider plots revealed the RMSE of deconvolution results of 18 methods among 6 cell types from MERFISH (100 μm , 50 μm and 20 μm resolution per spot).

In the second figure, we showed the JSD (top) and RMSE (bottom) results of all methods through all methods of all 36 samples in MERFISH datasets under three kinds of resolution (100 μm , 50 μm , 20 μm) and 12 samples. The larger and darker dots represent the higher RMSE or JSD.

Through the evaluation on zebrafish embryo datasets (the first figure), there were 3 datasets with different resolutions of spots represented by 3 different colors as the legend showed and we chose three cell types (Blood Vasculature, Notochord and YSL) for all datasets where each row represented the result of one cell type.

In the results of MERFISH datasets (the first two figures), the increasing of resolutions (100 μm, 50 μm, 20 μm) resulted in three kinds of spot numbers (3067, 13375, 44679) and a clear pattern both in two figures above showed that the performance of all methods tended to decrease through the increasing of spot resolutions. In the results of zebrafish embryo dataset, the PCCs tended to decrease through the

increasing of the resolutions of spots (15 μm , 10 μm , 5 μm) which had the same conclusion as the results from MERFISH datasets. The reason was that the smaller number of cells or larger number of cell types contained in one spot, the proportions of each kind of cell type in one spot would be closer to 0 or 1, especially in the spots with the resolution of 20 μm which would contain one cell only. In this circumstance, the predicted proportion would have more gap with ground-truth proportion resulting in the decreasing performance.

For the second question, you are right that cell types with higher abundance would affect the accuracy, especially raising the RMSE of individual cell types. Please see the following two figures. We used seqFISH+ (with 10000 genes) and MERFISH (spot resolution of 100 μm) as examples. We calculated the RMSE of all methods for each cell type and summed up the RMSE along with all methods to generate a vector whose length was the number of cell types. Then, we calculated the abundance of each cell type by summing up the cell-type proportions from the ground-truth spot-by-cell-type matrix along with spots and we also gathered a vector whose length was the number of cell types. Finally, we fitted these two vectors (RMSE of all cell types and abundance of all cell types) and plotted them as following two figures. Each dot in the following two figures represented one kind of cell type with its calculated RMSE (Y-axis) and abundance (X-axis) in all spots.

Through these two figures, we could clearly observe that there was a strong correlation between abundance and RMSE for each cell type which meant the abundance strongly affected the RMSE results in all cell types. To eliminate the effects of abundance of cell types, we adjusted the RMSE definition in the manuscript where we normalize the RMSE for each cell type by the abundance of each cell type (the sum of calculated proportions among all spots S_k).

$$RMSE = \sqrt{\frac{1}{K} \sum_{k=1}^K \frac{1}{S_k} \sum_{i=1}^I (P_{ik} - T_{ik})^2}$$

T_{ik} and P_{ik} represent the true and predicted proportion of cell type k in the i th spot through the number of total cell types K . We also supplied the files including both JSD score and RMSE values for individual cell types which could be useful for the users.

8. As a complement to evaluation of methods on real data, the authors might consider an evaluation on

simulated data where a ground truth is known. Although simulated data obviously cannot match all of the features of real data, the ground truth represents a gold-standard that is particularly important to draw convincing conclusions in benchmarking studies.

Thank you for the excellent suggestion. For the seqFISH+ and MERFISH datasets, we used them as the simulated datasets with ground truth to evaluate the performance of all methods. Because for each of them, we had the spatial coordinates (X and Y), vector of gene expression profiles and the cell-type annotation of every single cell, thus, we could use a square to bin them and treat one square as a simulated spot with ground-truth cell-type proportion which could be calculated by the proportion of annotated cells. And through the sliding of squares, we could gather many simulated spots until all of cells are binned. The detailed procedure was shown in “Preprocessing of datasets” (Methods in manuscript).

9. Perhaps my own lack of knowledge, it is not clear to me why the authors excluded novoSpaRc and SpaOTsc in the benchmarking. These two methods are capable of allocating cells to spatial positions and therefore have wide applications in general.

Thank you very much for pointing out these two methods that we did not include in the original benchmarking. We have now included them and benchmarked them in all the experiments. The results are updated accordingly in all figures and tables.

10. The evaluation of the usability (Figure 2, which should be a table instead) is insufficient. Why not all the datasets were assessed to compare the usability? Will the number of cell types and shared genes effect the usability? Could the authors clearly define the user-friendliness for each method? Why some are more “friendly” than the others?

Thank you for the questions. To make our evaluation of usability more comprehensive, we recorded the computational time on Slide-seq V2 datasets with two kinds of cell-type numbers (11 and 17) and three kinds of gene numbers (full, 16000 and 8000). The results are shown in Figure 2 and Supp. Table 4 (the first table). The reason why we used Slide-seq V2 was that this dataset contained the largest number of spots (53208 spots) which is the key factor affecting the efficiency of all methods. Thus, this dataset could be used as the stress test which, we believe, is the most appropriate to test the limits of all methods.

From the following table, we could observe that the number of cell types affected the computational time from method to method because different methods had different strategies to assign the cell types for predicted results. On the other hand, the number of genes truly affected the computational time a lot. It may result from the huge difference among file size of input spatial transcriptomics data where the file size could extremely affect the loading of input file and the calculation in the pipeline.

Gene number	Full		8000		16000	
Cell type number	17	11	17	11	17	11
Berglund	426min	426min	372min	372min	402min	402min
NMFReg	33min	31min	9min	9min	19min	19min
stereoscope	652min	654min	190min	178min	388min	373min

SpatialDWLS	78min	69min	83min	72min	78min	68min
DSTG	7min	7min	6min	6min	8min	7min
SPOTlight	13min	11min	12min	12min	14min	10min
RCTD	305min	200min	129min	85min	209min	136min
Tangram	41min	16min	14min	5min	29min	12min
DestVI	134min	123min	37min	31min	83min	72min
STRIDE	161min	86min	49min	22min	100min	59min
SpiceMix	268min	231min	145min	133min	192min	150min
STdeconvolve	129min	119min	61min	50min	81min	77min
SpatialDecon	910min	749min	537min	335min	715min	534min
Cell2location	658min	569min	465min	437min	533min	485min
SD²	25min	24min	23min	21min	27min	25min
CARD	16min	14min	12min	11min	13min	11min
novoSpaRc	357min	339min	298min	259min	337min	320min
SpaOTsc	210min	210min	186min	186min	198min	198min

As for “user-friendliness”, inspired by previous benchmarking study [6], we evaluated several important factors for readers and users, including document quality, code quality, installation procedure, compatibility for operating system (OS) and example analysis. By evaluating the richness or readability of these five factors, we scored these factors during our implementation of all computational methods. The scores ranged from 1 to 5 (1: worst, 5: best) and considered user-friendliness as the overall factor. The detailed results of scores were shown in the following table.

Methods	Document quality	Code quality	Installation procedures	Compatibility for OS	Example analysis
Berglund	3	5	5	3	3
NMFReg	3	3	5	5	3
stereoscope	5	5	5	4	4
SpatialDWLS	4	2	5	4	4
DSTG	3	4	4	4	1
SPOTlight	5	5	4	4	5
RCTD	4	5	5	5	5
Tangram	3	3	5	5	5
DestVI	5	3	4	5	5

STRIDE	5	4	5	4	5
SpiceMix	3	3	3	4	3
STdeconvolve	4	3	4	4	3
SpatialDecon	4	4	4	5	4
Cell2location	4	5	5	4	4
SD²	4	5	4	5	4
CARD	5	5	3	4	5
SpaOTsc	3	5	4	5	5
novoSpaRc	4	5	5	5	5

11. The readability of some figures, such as Fig3A, B and Supplementary Figure 6,7,8, are terrible. No corresponding metrics or statistics panels were included on the figures, making readers hard to follow.

Thank you for the suggestion. We have redrawn Figure 3A and B as you suggested (as the following figure). For Supplementary Figures 6, 7, and 8, we showed the detailed metrics of MERFISH (20 μ m, 50 μ m and 100 μ m) in Supplementary Figures 9 and 10 because the Supplementary Figure 6, 7, and 8 were too huge and the text would be too small for readers to look at.

Reference:

- [1] Q. Song and J. Su, “DSTG: deconvoluting spatial transcriptomics data through graph-based artificial intelligence,” *Brief. Bioinform.*, vol. 22, no. 5, p. bbaa414, Sep. 2021, doi: 10.1093/bib/bbaa414.
- [2] V. Kleshchevnikov *et al.*, “Cell2location maps fine-grained cell types in spatial transcriptomics,” *Nat. Biotechnol.*, 2022, doi: 10.1038/s41587-021-01139-4.
- [3] D. M. Cable *et al.*, “Robust decomposition of cell type mixtures in spatial transcriptomics,” *Nat. Biotechnol.*, 2021, doi: 10.1038/s41587-021-00830-w.
- [4] B. Tepe *et al.*, “Single-cell RNA-seq of mouse olfactory bulb reveals cellular heterogeneity and activity-dependent molecular census of adult-born neurons,” *Cell Rep.*, vol. 25, no. 10, pp. 2689–2703, 2018.
- [5] C. Liu *et al.*, “Spatiotemporal mapping of gene expression landscapes and developmental trajectories during zebrafish embryogenesis,” *Dev. Cell*, vol. 57, no. 10, pp. 1284–1298, 2022.
- [6] W. Saelens, R. Cannoodt, H. Todorov, and Y. Saeys, “A comparison of single-cell trajectory inference methods,” *Nat. Biotechnol.*, vol. 37, no. 5, pp. 547–554, 2019, doi: 10.1038/s41587-019-0071-9.

Reviewer #1 (Remarks to the Author):

The authors have properly and comprehensively responded to the reviewer comments. The revised manuscript is much better than the previous version. I am eager to see its published form which can have impacts and attract considerable citations in the trendy field of spatial transcriptomics.

Reviewer #2 (Remarks to the Author):

The authors have adequately addressed all of my comments.

Reviewer #3 (Remarks to the Author):

Reviewers' Comments:

I think the manuscript has been substantially revised, though some of my concerns in the first round of review still have not been fully addressed.

1. I asked the authors to claim the value and novelty of their study in Q#1 against the published papers, and the authors claimed the following in their response letter "In summary, compared with these two studies, other than the comprehensiveness of the methods we compared, the most significant novelty should be that we provided a full-scale summarization in Figure 2 for all methods over different datasets and various conditions, which offered readers and users a clear picture of these methods' performance landscape and guidance about which methods to choose and what parameters to set up in practice." I think it is not exactly true because the revised Figure 2 is very similar with Supplementary Table 5 of Li et al Nat. Methods, 2022 (PMID: 35577954).
2. The authors claimed the following in the revised manuscript "A recent benchmarking study [8] was focused on single-cell RNA sequencing (scRNA-seq) and spatial transcriptomics integration methods, and used only simulated data rather than real sequencing-based spatial transcriptomics data to evaluate the performance of various deconvolution methods." I think it is not exactly right because the benchmarking study the authors referred (Li et al Nat. Methods, 2022, PMID: 35577954) did use 32 real datasets to "simulate" the input data for ground truth. First of all, for single-cell resolution datasets such as seqFISH+ or MERFISH, the authors used binning to construct low-resolution data as ground truth, exactly the same way as Li et al Nat. Methods, 2022. The difference is Li et al called them as "simulated" datasets while the authors called them "real" datasets. Secondly, for data sets such as 10X Visium, which are inherently low-resolution, the authors used marker gene as the ground truth to annotate each cell type, which brings several new problems: (1) For some cell types the authors used one or two genes which will be greatly affected by sequencing noise, making the annotation of these cell types inaccurate; (2) For some cell types the authors used more than ten marker genes for cell type annotation, making the comparison between those cell types annotated by only one or two genes unfair.
3. In terms of normalization, several methods are better performed using raw data, such as DestVI, SpatialDecon, SPOTlight, RCTD, etc. while some others are better with normalized data. It is recommended that the authors make it clear in the revised manuscript.

Again, although a big chunk of novelty was taken by the two published papers earlier this year, I still think this is an important study. However, I hope that when the authors stating the uniqueness or advantages of their work, they should state the limitations of other published articles in line with the facts. I personally feel that the authors have overemphasized the use of different real data sets for benchmarking of deconvolution methods as one of the main innovations of their work.

We are very grateful to the three reviewers for their positive comments on our revision. Below please find the point-by-point response to all the reviewers' comments.

Reviewer #1 (Remarks to the Author):

The authors have properly and comprehensively responded to the reviewer comments. The revised manuscript is much better than the previous version. I am eager to see its published form which can have impacts and attract considerable citations in the trendy field of spatial transcriptomics.

Answer: Thank you very much for your support of our work.

Reviewer #2 (Remarks to the Author):

The authors have adequately addressed all of my comments.

Answer: Thank you very much for your support of our work.

Reviewer #3 (Remarks to the Author):

Reviewers' Comments:

I think the manuscript has been substantially revised, though some of my concerns in the first round of review still have not been fully addressed.

Answer: Thank you very much for your helpful and detailed comments. We have followed your comments to further revise our manuscript and we believe that the revision has addressed all your comments.

1. I asked the authors to claim the value and novelty of their study in Q#1 against the published papers, and the authors claimed the following in their response letter "In summary, compared with these two studies, other than the comprehensiveness of the methods we compared, the most significant novelty should be that we provided a full-scale summarization in Figure 2 for all methods over different datasets and various conditions, which offered readers and users a clear picture of these methods' performance landscape and guidance about which methods to choose and what parameters to set up in practice." I think it is not exactly true because the revised Figure 2 is very similar with Supplementary Table 5 of Li et al Nat. Methods, 2022 (PMID: 35577954).

Answer: Thanks for pointing this out. You are right that the summarized table in Figure 2 is quite similar to Supplementary Table 5 of Li et al Nat. Methods, 2022 (PMID: 35577954). Thus, we preferred to claim that our novelty against the Li et al Nat. Methods, 2022 (PMID: 35577954) should be two-fold: 1) much more compared methods; 2) clear guidelines and solid recommendations of method selection for users. We have further clarified these in the Discussion section.

2. The authors claimed the following in the revised manuscript "A recent benchmarking study [8] was focused on single-cell RNA sequencing (scRNA-seq) and spatial transcriptomics integration methods, and used only simulated data rather than real sequencing-based spatial transcriptomics data to evaluate the performance of various deconvolution methods." I think it is not exactly right because the benchmarking study the authors referred (Li et al Nat. Methods, 2022, PMID: 35577954) did use 32 real datasets to "simulate" the input data for ground truth. First of all, for single-cell resolution datasets such as seqFISH+ or MERFISH, the authors used binning to construct low-resolution data as ground truth, exactly the same way as Li et al Nat. Methods, 2022. The difference is Li et al called them as "simulated" datasets while the authors called them "real" datasets. Secondly, for data sets such as 10X Visium, which are inherently low-resolution, the authors used marker gene as the ground truth to annotate each cell type, which brings several new problems: (1) For some cell

types the authors used one or two genes which will be greatly affected by sequencing noise, making the annotation of these cell types inaccurate; (2) For some cell types the authors used more than ten marker genes for cell type annotation, making the comparison between those cell types annotated by only one or two genes unfair.

Answer: Thank you for the comment. We agree with you that a clearer distinction between “simulated” and “real” datasets is needed. We have thus revised the statement about the limitations of previous two studies in the manuscript (the third paragraph in the introduction), to make it fair for the previous works.

For the first point you raised, we admitted that there would be some noise in the correlation between marker genes and cell types. But the choice of marker genes was referred from previous studies (Supp. Table 2) which showed the credibility of strong correlation between these cell types and the corresponding marker genes. The experiments evaluated by multiple cell types in each dataset (Supp. Figure 11, 19) also proved that the selected marker genes in these cell types were sufficient to show the clear patterns of the performance of all methods, where the noise would not affect the performance and conclusion. For the second problem you raised, it is true that different numbers of marker genes were chosen in different cell types which could result in different PCCs. However, the main goal of these evaluations is not to compare the performance of all methods among different cell types, but rather on the performance of all methods under each cell type which means the key is the comparison of different methods instead of cell types.

3. In terms of normalization, several methods are better performed using raw data, such as DestVI, SpatialDecon, SPOTlight, RCTD, etc. while some others are better with normalized data. It is recommended that the authors make it clear in the revised manuscript.

Answer: Thank you for pointing this out and we have followed your comment to clarify this point more in the revision (in the “Robustness” part). For many methods (e.g., DestVI, SpatialDecon and SPOTlight), they have the normalization procedures in their own pipelines and input data could be normalized by default. Thus, the inputted normalized data has a duplicate normalization than raw data which results in the worse performance than when inputting with raw data. For the other methods (e.g., SpaOTsc and Tangram), they do not have default normalization procedures in their pipelines which would cause better performance when inputting normalized data than raw data.